# RAINDROP GS: A BENCHMARK FOR 3D GAUSSIAN SPLATTING UNDER RAINDROP CONDITIONS

## ABSTRACT

3D Gaussian Splatting (3DGS) under raindrop conditions suffers from severe occlusions and optical distortions caused by raindrop contamination on the camera lens, substantially degrading reconstruction quality. Existing benchmarks typically evaluate 3DGS using synthetic raindrop images with known camera poses (constrained images), assuming ideal conditions. However, in real-world scenarios, raindrops often interfere with accurate camera pose estimation and point cloud initialization. Moreover, a significant domain gap between synthetic and real raindrops further impairs generalization. To tackle these issues, we introduce **RaindropGS**, a comprehensive benchmark designed to evaluate the full 3DGS pipeline—from unconstrained, raindrop-corrupted images to clear 3DGS reconstructions. Specifically, the whole benchmark pipeline consists of three parts: data preparation, data processing, and raindrop-aware 3DGS evaluation, including types of raindrop interference, camera pose estimation and point cloud initialization, single image rain removal comparison, and 3D Gaussian training comparison. First, we collect a real-world raindrop reconstruction dataset, in which each scene contains three aligned image sets: raindrop-focused, background-focused, and rain-free ground truth, enabling a comprehensive evaluation of reconstruction quality under different focus conditions. Through comprehensive experiments and analyses, we reveal critical insights into the performance limitations of existing 3DGS methods on unconstrained raindrop images and the varying impact of different pipeline components: the impact of camera focus position on 3DGS reconstruction performance, and the interference caused by inaccurate pose and point cloud initialization on reconstruction. These insights establish clear directions for developing more robust 3DGS methods under raindrop conditions.

## 1 INTRODUCTION

3D Gaussian Splatting (3DGS) in raindrop-contaminated scenes presents significant challenges, as adherent raindrops on camera lenses cause severe occlusions and optical distortions Li et al. (2024); Liu et al. (2025); Qian et al. (2024). These artifacts disrupt image correspondence, degrade the quality of camera pose estimation and point cloud initialization Zhu et al. (2023), both of which are essential for successful 3DGS reconstruction. Moreover, the presence of raindrops varies across views, blurring images by changing the camera focal plane You et al. (2013), introducing multi-view inconsistencies that further hinder reconstruction fidelity Petrovska & Jutzi (2025).

Several recent methods Li et al. (2024); Liu et al. (2025); Qian et al. (2024) have explored 3D Gaussian Splatting under raindrop scenarios and demonstrated promising results on synthetic datasets. However, such evaluation settings are overly idealized and fail to capture the complexity and diversity of real-world conditions. To be specific, these methods typically assume the raindrop inputs are constrained images, where a clear details of both raindrops shape and background scenes, a good camera pose and point cloud initialization. However, acquiring such information from real-world raindrop-affected images is challenging Huang et al. (2025); Zhang et al. (2024). Inaccuracies in pose estimation and point cloud initialization can significantly degrade the quality of subsequent 3DGS reconstruction Wang et al. (2024); Fu et al. (2024). Furthermore, the substantial domain gap between synthetic and real raindrops raises concerns about generalization. Methods validated on synthetic datasets often fail to perform well when applied to real-world scenes. As illustrated in Fig-

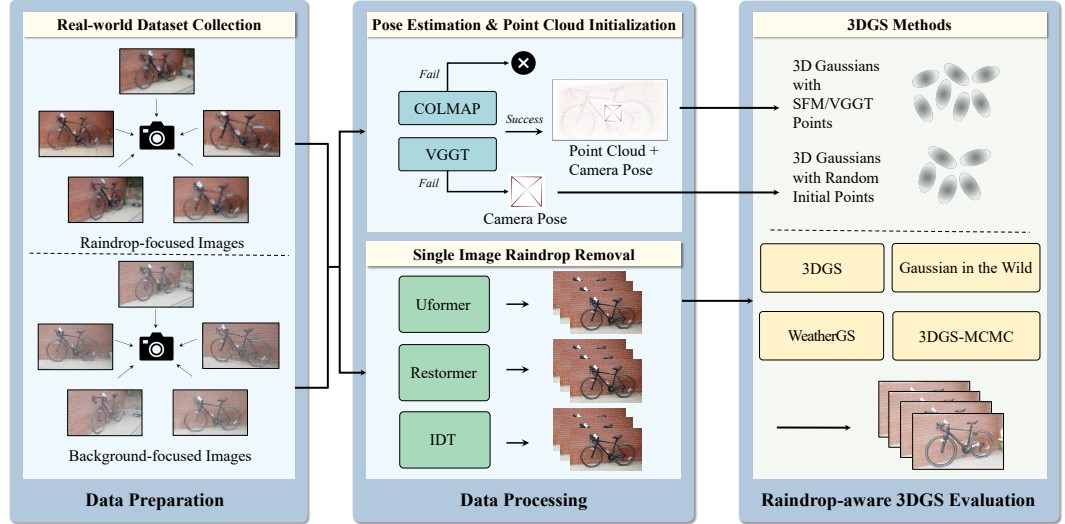

Figure 1: 3DGS Raindrop Reconstruction Benchmark Pipeline. We develop the first benchmark for comprehensively evaluating 3DGS performance under raindrop conditions. The benchmark begins with real-world dataset collection, proceeds through data processing, and ends with a raindrop-aware 3DGS evaluation. In particular, we assess how raindrop-induced image contamination reduces the number of points available for cloud initialization and degrades camera pose estimation, and how these factors impact the performance of 3DGS methods.

ure 2 (a-c), these synthetic datasets are valuable but exhibit limitations, such as the same raindrop shape and position across different views.

To address these issues, we introduce **RaindropGS**, a comprehensive benchmark for evaluating the complete raindrop 3DGS pipeline—from unconstrained, raindrop-corrupted input images to clear 3DGS reconstructions. Specifically, our pipeline consists of three stages: data preparation, data processing, and raindrop-aware 3DGS evaluation. For data preparation, we compare the effects of different types of images (raindrop-focused and background-focused) on the subsequent reconstruction process. During data processing, we evaluate the performance of camera pose estimation and point cloud initialization, as well as single-image raindrop removal algorithms. In raindrop-aware 3DGS evaluation, we consider methods that may affect the performance of real-world raindrop reconstruction, such as raindrops and point cloud Gaussian optimization.

In addition, we collect a real-world 3D reconstruction dataset captured under raindrop conditions. For each scene, three aligned image sets are acquired: raindrop-focused, background-focused, and rain-free ground truth. This design enables evaluation of the full pipeline in real-world scenarios as well as under different focus conditions. As shown in Figure 2 (d), our RaindropGS dataset reflects real-world conditions, featuring multiple focus settings and a diverse range of raindrop characteristics.

Using the collected dataset, we process the images (both raindrop-focused and background-focused) to obtain the corresponding rain-free images, estimated camera pose, and initialized point cloud. To analyze the impact of raindrops on the real-world dataset collection, we use COLMAP Schonberger & Frahm (2016b) and VGGT Wang et al. (2025) to estimate the camera pose and initialize the point cloud, enabling us to investigate how sequence-based and feed-forward approaches influence the performance of 3DGS methods. We include three widely used deraining methods, Uformer Wang et al. (2022), Restormer Zamir et al. (2022), and IDT Xiao et al. (2022) in the raindrop removal stage, comparing the impact of different raindrop removal methods on subsequent 3DGS reconstruction performance. For the raindrop-aware 3DGS evaluation, we integrate multiple 3DGS variants, including the original 3DGS Kerbl et al. (2023), WeatherGS Qian et al. (2024), GS-W Zhang et al. (2024), and 3DGS-MCMC Kheradmand et al. (2024), to evaluate the impact of different reconstruction strategies on raindrop-corrupted inputs. These methods are evaluated under varying pre-processing pipelines and focus conditions to assess their robustness and adaptability.

Through rigorous quantitative and qualitative analyses, we evaluate the performance of state-of-the-art 3DGS methods under raindrop conditions, as well as their pre-processing stages. The results revealing their strengths, limitations, and sensitivity to different pre-processing and focus settings. These findings not only benchmark the current progress but also highlight key challenges and future directions for improving 3DGS performance in real-world adverse environments. Our main contributions are summarized as follows:

- **Full 3DGS Pipeline Benchmark:** We introduce the first 3DGS benchmark for raindrop-contaminated scenes, covering the complete pipeline from unconstrained, raindrop-corrupted images to the final 3D Gaussian reconstructions.

- **First Real-world Dataset:** We collect a real-world 3DGS raindrop reconstruction dataset with aligned raindrop-focused, background-focused, and rain-free ground truth images, enabling comprehensive evaluation of reconstruction quality across different focus conditions.

- **Comparative Study and Insights:** We validate existing 3DGS methods on our benchmark, revealing their strengths and limitations, and providing insights into future research directions.

## 2 RELATED WORK

**3DGS Reconstruction under Raindrop Conditions**   In recent years, 3DGS has emerged as a powerful technique for scene reconstruction. Unlike NeRF Mildenhall et al. (2021), it represents scenes using a sparse set of 3D Gaussians, enabling real-time rendering. However, standard 3DGS benchmarks assume clear input views, and performance often degrades when images contain transient occlusions such as raindrops on the lens Liu et al. (2025); Qian et al. (2024); Kulhanek et al. (2024).

To address this issue, several methods  Li et al. (2024); Qian et al. (2024); Liu et al. (2025) have been developed to improve 3D reconstruction in raindrop scenes. WeatherGS Qian et al. (2024) first generates raindrop masks to identify occluded regions and then reconstructs clear scenes by excluding these areas during 3D Gaussian Splatting. Meanwhile, DerainGS Liu et al. (2025) incorporates a dedicated image enhancement module to remove raindrop artifacts and employs supervised Gaussian-ellipsoid fitting, achieving 3D deraining in the final output. These methods are trained on synthetic raindrops and deliver strong results under the assumption of accurate camera pose estimation and reliable point cloud initialization. However, they overlook the initial disruptions that real raindrops introduce to both pose estimation and point cloud initialization, resulting in poor generalization to real-world raindrop scenarios.

**Raindrop Removal Methods**   To mitigate lens occlusion artifacts, single-image derain methods have been extensively studied. Early works such as Raindrop Removal Network Qian et al. (2018) leverage visual attention to segment and inpaint raindrop regions, while UMAN Shao et al. (2021) extends this idea with multiscale feature fusion. More recently, transformer-based restoration models (for example, Restormer Zamir et al. (2022), Uformer Wang et al. (2022) , DiT Peebles & Xie (2023) and IDT Xiao et al. (2022)) demonstrate superior restoration under heavy rainfall by modeling long range dependencies. However, these methods process each image independently and do not enforce cross-view consistency, leading to reconstruction artifacts when applied as a preprocessing step for 3D reconstruction.

**3D Raindrop Reconstruction Benchmark and Dataset**   Current 3DGS raindrop reconstruction methods focus primarily on the Gaussian fitting stage and ignore the influence of earlier steps on the training process, such as camera pose estimation and point cloud initialization. In addition, they rely on synthetic training datasets created by Blender on clear images Liu et al. (2025); Li et al. (2024), which creates a significant domain gap and prevents accurate evaluation in real-world conditions. A few real-world datasets have tried to simulate rain on camera lenses for stereo or small scale multi view setups. DerainNeRF Li et al. (2024) captures stereo pairs by spraying water onto a glass plate in front of a calibrated rig and provides binary raindrop masks. WeatherGS Qian et al. (2024) extracts key frames from publicly available rainy videos but does not supply a ground truth reference. Overall, existing datasets remain mostly synthetic and do not reflect real-world

Figure 2: Example of existing raindrop 3D datasets (DerainNeRF Li et al. (2024), WeatherGS Qian et al. (2024), DerainGS Liu et al. (2025)) and our RaindropGS Dataset. As indicated by the red boxes, existing datasets exhibit the same raindrop distribution across different viewpoints; in contrast, the green boxes illustrate the diversity of raindrop distributions in our dataset. For each viewpoint, we include both raindrop-focused and background-focused images and provide corresponding clear images for 3DGS performance evaluation.

raindrop interference, and current algorithms overlook the early stages of the pipeline, making their performance evaluation under real conditions unreliable.

To address this challenge, we revisit the complete 3DGS raindrop reconstruction pipeline and develop a benchmark covering every stage: data preparation, data processing, and raindrop-aware 3DGS evaluation. To evaluate current algorithms and guide future research, we compile a real-world dataset of eleven scenes.

## 3 RAINDROPGS BENCHMARK AND DATASET

In this section, we describe in detail the components of the 3DGS raindrop reconstruction benchmark, which consists of three parts. The first part describes the specific data collection process. The second and third parts describe the selection and evaluation of different models.

### 3.1 DATA PREPARATION

In this section, we first describe our data collection process, including the underlying optical refraction model and acquisition setup. We also present dataset statistics and comparisons with existing datasets.

**Data Collection**    To begin with, we consider a pinhole camera model focused on the background plane. In the absence of optical distortion (e.g., caused by raindrops), all scene elements located on the focal plane would appear sharp and well-defined. However, raindrops adhering to a thin cover glass placed directly in front of the lens act as miniature convex lenses, introducing optical distortion and causing defocus. When background rays intersect a raindrop, they are refracted at the curved surface of the drop decided by Snell Law Born & Wolf (2013). In contrast, rays that do not encounter any raindrop travel without deviation through the imaging system to the sensor. Consequently, refracted and non-refracted rays map to spatially distinct locations on the image plane, illustrating how the presence of raindrops directly affects the imaging distortion. Furthermore, since raindrops

Table 1: Comparative raindrop 3D Reconstruction datasets. Compared to existing collections, our dataset spans a greater variety of scenes and distinguishes between raindrop-focused and background-focused captures.

| Dataset | Scene count | | Images (Real) | GT (Real) | Camera focus | |
| --- | --- | --- | --- | --- | --- | --- |
| | Real | Synthetic | | | Raindrop | Background |
| DerainNeRF | 3 | 2 | 20–25 | × | × | ✓ |
| DerainGS | 7 | 6 | 22–35 | × | × | ✓ |
| **RaindropGS** | **11** | × | **24–53** | ✓ | ✓ | ✓ |

don't fully transmit light, the regions under raindrops exhibit localized intensity attenuation. This attenuation produces visible artifacts.

By contrast, another alternative configuration in which the camera is set to focus on the raindrop plane rather than the background plane. In this configuration, the image plane captures sharp, in-focus representations of the raindrop surfaces. Under these circumstances, more distant background features, seen through each raindrop, appear as miniaturized projections and are blurred in areas outside the raindrops.

To create the dataset, we use a pan-tilt sphere platform to keep the camera stationary. We then follow a standardized protocol grounded in optical refraction principles to ensure consistent camera alignment while allowing raindrops to vary in location, shape and size. The setup consists of two professional tripods with ball heads, a calibrated pressure sprayer, and a glass plate with over 98 percent light transmittance. The glass plate is tilted at an angle between $0°$ and $30°$ from the vertical with respect to the ground and is placed approximately 3 centimeters in front of the camera lens.

**Data Statistics**   We summarize our dataset in Table 1. The dataset includes 11 real-world scenes, each containing 24 to 53 images captured under unconstrained raindrop conditions. For every viewpoint, three aligned images are provided: a raindrop-focused image, a background-focused image, and a clean ground-truth image. The raindrops in each viewpoint vary randomly in shape, number, and size, closely replicating real-world conditions. In contrast, existing synthetic datasets for 3DGS lack representation of camera focus effects on raindrop images and do not include diverse raindrop appearances across multiple viewpoints.

**Focus Shift**   During image capture (Figure 2(d)), raindrops adhering to the front glass shift the camera's focal plane. When many raindrops lie within the depth of field, the camera focuses on them and the background becomes blurred. Conversely, if only a few raindrops fall within the focal region, the camera focuses on the background and the raindrops appear out of focus. Most synthetic datasets ignore focus variation and render both background and raindrops as sharply in focus, which may reduce 3DGS reconstruction accuracy on real images. The RaindropGS dataset explicitly addresses this issue by capturing each scene under both raindrop-focused and background-focused conditions to support more realistic 3DGS raindrop evaluation.

## 3.2   DATA PROCESSING

Our data processing pipeline consists of two main components: pose estimation and point cloud initialization, and single-image raindrop removal pre-processing. Unlike existing raindrop Gaussian splatting methods that assume known camera poses and accurate point clouds, our benchmark directly estimates both the camera poses and an initial point cloud from the raindrop-affected images. This approach enables us to evaluate the robustness of the subsequent 3DGS reconstruction against potential errors in pose estimation and inaccuracies in the initial point cloud. To obtain a clean 3DGS reconstruction in the raindrop-aware 3DGS evaluation stage, we apply raindrop removal techniques to the multi-view raindrop images.

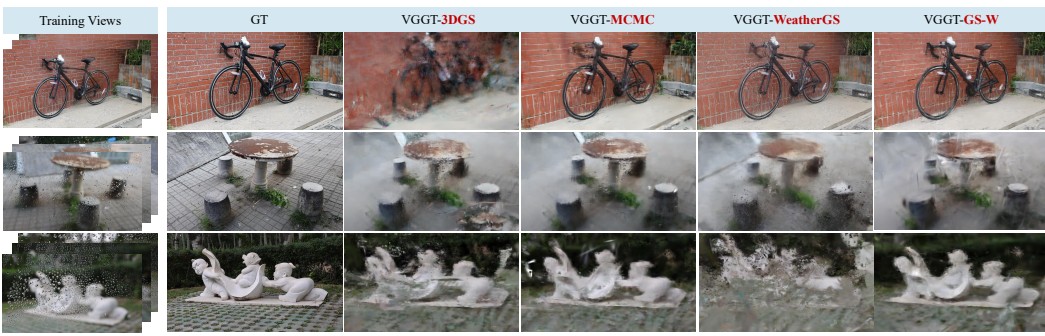

Figure 3: Qualitative Comparison among 3DGS methods: On the raindrop-focused dataset, the original 3DGS loses structural integrity, while GS-W and 3DGS-MCMC retains scene completeness. WeatherGS faithfully reconstructs the scene after raindrop removal but fails to correct background blur. On the background-focused dataset, all methods show moderate performance with artifacts.

**Pose Estimation and Point Cloud Initialization**    To estimate the camera pose and initialize the point cloud from multi-view raindrop images, we employ COLMAP Schonberger & Frahm (2016b) and Visual Geometry Grounded Transformer (VGGT) Wang et al. (2025).

COLMAP is a robust tool capable of performing both Structure-from-Motion (SfM) Schonberger & Frahm (2016a) and Multi-View Stereo (MVS) Furukawa et al. (2015). We leverage SfM to estimate intrinsic and extrinsic camera parameters and MVS to generate the initial point cloud. However, raindrop interference often impedes reliable feature matching across viewpoints. This results in significant errors in estimated camera parameters and a drastic reduction in initialized point cloud density. To overcome the limitations of SfM, we employ VGGT as a comparative baseline. VGGT, a feed-forward unified method for pose estimation and point cloud generation, is more robust to raindrop interference due to its use of DINO.

In raindrop-focused scenes, the background is often too blurred for reliable scene initialization, causing both COLMAP Schonberger & Frahm (2016b) and VGGT Wang et al. (2025) to fail. In certain scenes, COLMAP may suffer a substantial reduction in the number of matchable camera poses due to degraded Correspondence Search Schonberger & Frahm (2016b) performance, which ultimately leads to reconstruction failure. To address cases where raindrop interference and blur reduce the initial point cloud produced by COLMAP and VGGT, we employ a random point-cloud initialization strategy. Specifically, 100,000 points are randomly initialized, matching the order of magnitude of point counts obtained from ground-truth scenes.

**Raindrop Removal**    Since traditional 3DGS methods do not incorporate raindrop removal capabilities, we employ three widely used single-image restoration models. Uformer Wang et al. (2022) applies non-overlapping window-based self-attention and a multi-scale restoration modulator, demonstrating superior capability in restoring details from raindrop-affected and blurry images. Restormer Zamir et al. (2022) leverages multi-Dconv head transposed attention and a gated-Dconv feed-forward network to restore high-quality images, while IDT Xiao et al. (2022) employs a dual Transformer with window- and spatial-based designs for rain streak and raindrop removal.

All raindrop removal methods are trained on the Raindrop Clarity dataset Jin et al. (2024) to acquire raindrop removal capabilities. Raindrop Clarity is a dataset containing both daytime and nighttime image pairs, though we only use the daytime data for training. Furthermore, Raindrop Clarity includes both background-focused and raindrop-focused image pairs, making it well-suited for our task.

### 3.3 RAINDROP-AWARE 3DGS EVALUATION

With the estimated camera poses and initialized point cloud, we proceed to evaluate four representative 3DGS methods: 3DGS Kerbl et al. (2023), WeatherGS Qian et al. (2024), GS-W Zhang et al. (2024), and 3DGS-MCMC Kheradmand et al. (2024).

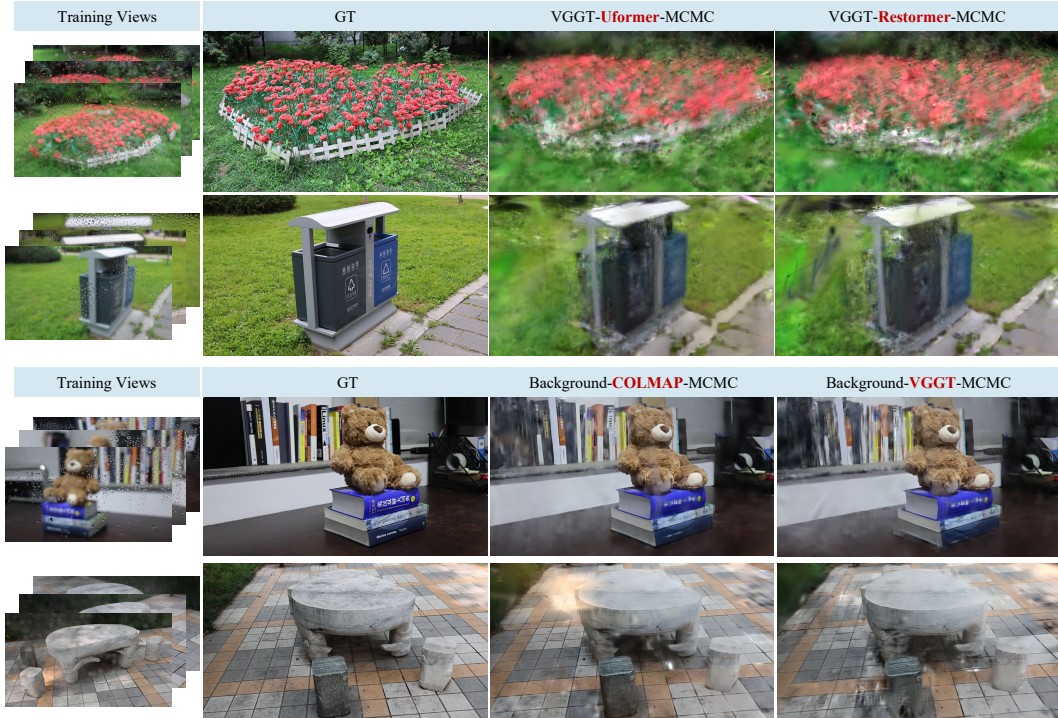

Figure 4: Qualitative Comparison across data pre-processing: on the raindrop focused dataset, the original 3DGS method loses the structural integrity of the reconstruction, whereas GS-w and 3DGS-MCMC retain scene completeness to some extent. WeatherGS most faithfully reproduces the reconstruction after raindrop removal but fails to correct background blur. On the background focused dataset, all methods perform at a moderate level and exhibit residual white haze, consistent with our quantitative analysis.

Among these, 3DGS Kerbl et al. (2023) serves as a standard baseline for 3D Gaussian splatting. WeatherGS Qian et al. (2024) incorporates single-image raindrop removal, so we omit the explicit raindrop removal step in its data processing pipeline. GS-W Zhang et al. (2024) is specifically designed for challenging conditions and unconstrained image collections, making it more robust to inconsistent multi-view inputs. 3DGS-MCMC Kheradmand et al. (2024), on the other hand, does not rely on accurate point cloud initialization. Each of the aforementioned methods has its own advantages, making their evaluation in our benchmark both meaningful and insightful.

# 4 EXPERIMENTS

## 4.1 IMPLEMENTATION DETAILS

We standardize all scene images to a resolution of 1024 * 576 for uniform comparison. During VGGT-based camera pose estimation and point cloud initialization, we follow the authors' pre-processing protocol, resizing each input image to 518 * 518 before processing. Likewise, for Uformer Wang et al. (2022) and Restormer Zamir et al. (2022), we downscale images to 256 * 256, and then tile the results to restore the original resolution.

For all 3DGS methods, we adhere to their official optimization settings, we run 30,000 iterations for 3DGS, WeatherGS, and 3DGS-MCMC, and 70,000 iterations for GS-W.

For 3DGS-MCMC, we set the initial point cloud size to 100,000, based on the number of points that VGGT can initialize in our dataset. All models are implemented in PyTorch and trained concurrently on 8 NVIDIA RTX 3090 GPUs.

Table 2: Comparison of camera pose estimation (AUC@30 Wang et al. (2023)) and the number of points in the point cloud between VGGT and COLMAP on background- and raindrop-focused datasets.

| | VGGT | | COLMAP | |
|---|---|---|---|---|
| | BG-focused | RD-focused | BG-focused | RD-focused |
| AUC@30 | 0.91 | 0.34 | 0.79 | 0.17 |
| 3D points | 69401.11 | 0 | 5476.89 | 302.50 |

Table 3: Comparison of the performance of different deraining methods. Uformer Wang et al. (2022) achieves the best performance, Restormer Zamir et al. (2022) attains performance that closely matches Uformer's.

| PSNR ↑ | | | SSIM ↑ | | | LPIPS ↓ | | |
|---|---|---|---|---|---|---|---|---|
| IDT | Uformer | Restormer | IDT | Uformer | Restormer | IDT | Uformer | Restormer |
| 22.025 | 26.731 | 26.249 | 0.623 | 0.808 | 0.784 | 0.251 | 0.162 | 0.171 |

## 4.2 QUANTITATIVE COMPARISON

Table 4 compares the impact of background-focused (BG-focused) and raindrop-focused (RD-focused) captures on 3DGS performance using VGGT Wang et al. (2025). For the original 3DGS method, raindrop-focused images exhibit a 4 dB drop compared to background-focused images, due to background blur and light refraction. VGGT processes all scenes but generates a point cloud with 0 points for raindrop-focused images, for which we use a randomly initialized point cloud.

Table 2 and Table 5 compares the performance of VGGT Wang et al. (2025) and COLMAP Schonberger & Frahm (2016b). For camera pose estimation, we use VGGT and COLMAP to estimate poses on the ground-truth images and compare these estimates with the poses obtained from the corresponding images. VGGT yields more accurate camera pose estimates; both methods accurately recover poses for background-focused images but exhibit substantial performance degradation on raindrop-focused images. In terms of point cloud initialization, VGGT outperforms COLMAP for background point clouds, yet it fails to initialize the raindrop-focused dataset. However, when COLMAP successfully produces camera poses and an initialized point cloud, the original 3DGS method achieves the best performance with Uformer Wang et al. (2022).

Table 3 reports the performance of different deraining/restoration methods on raindrop-affected scenes. Uformer Wang et al. (2022) and Restormer Zamir et al. (2022) show comparable results, with Uformer Wang et al. (2022) slightly outperforming Restormer Zamir et al. (2022) in reconstruction metrics, while IDT Xiao et al. (2022) lags substantially behind the other two methods.

For 3DGS methods, GS-W Zhang et al. (2024) with VGGT Wang et al. (2025) and Uformer Wang et al. (2022) preprocessing achieves the best performance (PSNR = 19.123), due to its adaptive optimization strategy for handling occlusions and environmental variations in outdoor scenes. The second best is 3DGS-MCMC Kheradmand et al. (2024) (PSNR = 18.239) on background-focused scenes with IDT Xiao et al. (2022) and VGGT, which shows robustness to initialization.

## 4.3 QUALITATIVE COMPARISON

Figures 3 and 4 show the qualitative results of 3DGS Kerbl et al. (2023), WeatherGS Qian et al. (2024), GS-W Zhang et al. (2024), and 3DGS-MCMC Kheradmand et al. (2024), along with their Uformer Wang et al. (2022) and Restormer Zamir et al. (2022) outputs. Scenes with background-focused images exhibit generally good performance, while raindrop-focused images pose significant reconstruction challenges due to the loss of background details and the presence of undetected raindrops. Uformer and Restormer outperform WeatherGS in restoring raindrop-degraded images. Among 3DGS variants, 3DGS suffers from detail loss, GS-W introduces artifacts, and 3DGS-MCMC offers improved quality over 3DGS. Weather-GS exhibits considerable blurriness

Table 4: The quantitative evaluation of baseline approaches on the RaindropGS dataset. With VGGT and Uformer preprocessing, GS-W achieves the best performence. These 3DGS variants excel on background-focused dataset but show significantly lower performance on raindrop-focused dataset.

| Focus | Metrics | 3DGS | | | 3DGS-MCMC | | | GS-W | | | WeatherGS |
|---|---|---|---|---|---|---|---|---|---|---|---|
| | | Uformer | Restormer | IDT | Uformer | Restormer | IDT | Uformer | Restormer | IDT | / |
| RD-focused | PSNR↑ | 13.894 | 13.876 | 13.958 | 15.109 | 15.005 | 14.994 | 16.099 | 15.400 | 15.873 | 13.070 |
| | SSIM ↑ | 0.346 | 0.345 | 0.350 | 0.383 | 0.380 | 0.383 | 0.512 | 0.484 | 0.511 | 0.307 |
| | LPIPS ↓ | 0.657 | 0.653 | 0.658 | 0.654 | 0.649 | 0.659 | 0.808 | 0.828 | 0.798 | 0.658 |
| BG-focused | PSNR ↑ | 17.906 | 17.741 | 18.094 | 18.219 | 18.148 | 18.239 | 19.123 | 19.074 | 17.818 | 17.124 |
| | SSIM ↑ | 0.478 | 0.469 | 0.480 | 0.482 | 0.477 | 0.483 | 0.555 | 0.550 | 0.507 | 0.428 |
| | LPIPS ↓ | 0.459 | 0.455 | 0.438 | 0.486 | 0.482 | 0.478 | 0.483 | 0.479 | 0.526 | 0.436 |

Table 5: Comparison of COLMAP and VGGT on RaindropGS background-focused datasets with Uformer. Owing to unsuccessful camera pose estimation by COLMAP, 3DGS methods could not be applied to the raindrop-focused datasets.

| | 3DGS | | 3DGS-MCMC | | GS-W | | WeatherGS | |
|---|---|---|---|---|---|---|---|---|
| | COLMAP | VGGT | COLMAP | VGGT | COLMAP | VGGT | COLMAP | VGGT |
| PSNR ↑ | 19.512 | 18.167 | 17.919 | 18.248 | 18.339 | 20.033 | 18.094 | 17.099 |
| SSIM ↑ | 0.603 | 0.473 | 0.492 | 0.474 | 0.588 | 0.613 | 0.504 | 0.428 |
| LPIPS ↓ | 0.384 | 0.467 | 0.454 | 0.489 | 0.562 | 0.433 | 0.400 | 0.447 |

due to limitations in handling raindrop-degraded images but shows strong multi-view consistency. In scenes that both COLMAP and VGGT successfully reconstruct, COLMAP recovers finer geometric and photometric detail. However, under raindrop interference, COLMAP fails to reconstruct any raindrop-focused scenes from its estimated camera poses and point clouds. By contrast, VGGT demonstrates superior robustness.

## 4.4 DISCUSSION

Raindrop-focused and background-focused images perform distinctly in synthetic benchmarks. Raindrop-focused images suffer significant reconstruction quality loss due to the absence of background detail, highlighting deficiencies in current raindrop feature modeling and background separation. These results reveal a substantial domain gap between synthetic environments and real-world raindrop conditions, suggesting the need for future frameworks that integrate raindrop scattering models and adaptive information completion to improve performance in real precipitation scenarios. In camera pose estimation and point cloud initialization, more robust models will stabilize reconstruction under raindrop conditions, as current methods struggle with point cloud initialization when images are focused on raindrops. 3DGS methods that can effectively handle occlusions and illumination variations will have a clear advantage. Additionally, developing better training strategies for point cloud initialization is a key area for future research.

## 5 CONCLUSION

In summary, RaindropGS offers a novel benchmark for evaluating 3DGS methods under real-world raindrop conditions. By addressing the limitations of previous synthetic datasets, we provide a more accurate representation of 3DGS performance in practical, unconstrained environments. Through the evaluation of multiple 3DGS variants, we identify the accumulated errors through camera pose estimation, point cloud initialization, raindrop removal, and 3DGS methods. Our findings highlight the strengths and weaknesses of existing approaches, offering insights into their performance under raindrop-corrupted conditions. These results underscore the need for more robust techniques to handle diverse raindrop characteristics and multi-view inconsistencies. RaindropGS not only contributes to the advancement of 3D reconstruction under challenging conditions but also lays the foundation for future research aimed at improving 3DGS performance in real-world applications.

## REPRODUCIBILITY STATEMENT

We captured a real-world dataset for raindrop-affected 3D Gaussian Splatting (3DGS) reconstruction. During acquisition, we recorded 2K-resolution video using a Canon R8 paired with an RF 24–50 mm lens. Randomly sprayed water droplets were applied to a 2 mm thick glass plate with optical transmittance exceeding 98%, which served as the raindrop-bearing surface. The camera and glass plate were rigidly mounted on commercially available tripods and kept stable throughout capture. After recording, keyframes were extracted from the videos to construct the dataset. For the benchmark comparisons, all code and implementations were obtained from publicly accessible repositories and websites; the entire procedure is reproducible.

## ETHICS STATEMENT

All authors of this paper have read and agreed to abide by the Code of Ethics.

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

# A APPENDIX

## A.1 USE OF LARGE LANGUAGE MODELS (LLMS)

The authors acknowledge the use of large language models (LLMs) for phrasing polishing and grammar correction during manuscript preparation.

## A.2 DATASET OVERVIEW

Table 6 presents the statistics of our dataset, detailing the number of images under different conditions (raindrop-focused, background-focused, and ground truth) for each scene. In total, we captured 1,018 images across 11 distinct scenes.

## A.3 TECHNICAL DETAILS

Our dataset was collected using the video recording function of a fixed camera setup. The camera and glass plate were kept stationary, while the water droplet spray was adjusted to vary the density, position, and shape of the raindrops. Additionally, we modified the angle of the glass plate relative to the vertical plane to mitigate potential reflective effects.

All images were initially captured at 2K resolution. For evaluation, we uniformly downsampled the images to 1024 * 576 resolution. For single-frame raindrop removal using Uformer Wang et al. (2022) and Restormer Zamir et al. (2022), we apply full-resolution stitching to preserve spatial resolution. Similarly, during 3DGS reconstruction, we maintained the same resolution to ensure high-fidelity geometric consistency.

3DGS Kerbl et al. (2023), WeatherGS Qian et al. (2024), and 3DGS-MCMC Kheradmand et al. (2024) share similar reconstruction and rendering times, each requiring approximately 15–20 minutes on our experimental setup, with training conducted over 30,000 iterations. In contrast, GS-W Zhang et al. (2024) is officially recommended to be trained for 70,000 iterations, with total training and rendering times ranging from 150 to 180 minutes. Although GS-W delivers strong performance, its main limitation lies in its significantly longer training time.

## A.4 FURTHER EXPERIMENTAL RESULTS

We conduct a thorough comparison of the 3DGS pipeline across different scenes. Table 8 summarizes the quantitative performance of 3DGS Kerbl et al. (2023), WeatherGS Qian et al. (2024), GS-W Zhang et al. (2024), and 3DGS-MCMC Kheradmand et al. (2024). On the raindrop-focused dataset, 3DGS-MCMC Kheradmand et al. (2024) achieves the highest performance on the rusty-desk scene, with a PSNR of 18.572, followed closely by GS-W Zhang et al. (2024) with a PSNR of 18.463. On the background-focused dataset, GS-W Zhang et al. (2024) demonstrates the best performance on the same scene, reaching a PSNR of 21.688. Qualitative analysis in Figure 6 also shows this conclusion. These results suggest that different types of raindrop occlusions may require distinct processing strategies for optimal performance.

In Table 9, we compare two of three state-of-the-art single-image raindrop removal algorithms: Uformer Wang et al. (2022) and Restormer Zamir et al. (2022). Uformer Wang et al. (2022) demonstrates slightly better performance across the evaluated metrics. In the qualitative analysis shown in Figure 5, it can be seen that Uformer Wang et al. (2022) and Restormer Zamir et al. (2022) have similar raindrop removal capabilities, but neither can completely remove raindrops. On the raindrop-focused dataset, they also introduce additional artifacts.

In Table 7, we compare the impact of COLMAP Schonberger & Frahm (2016b) and VGGT Wang et al. (2025) on the performance of 3DGS reconstruction in the presence of raindrop degradation, using Uformer Wang et al. (2022) for single-image restoration. COLMAP Schonberger & Frahm (2016b) achieves superior performance when it successfully completes point cloud initialization and camera pose estimation. However, COLMAP Schonberger & Frahm (2016b) fails to perform these steps across the entire raindrop-focused dataset. Furthermore, several scenes in the background-focused dataset also encounter initialization failures, resulting in the breakdown of subsequent 3DGS methods reconstruction. In the qualitative analysis shown in Figure 7, COLMAP Schonberger &

Table 6: Summary of image counts in our dataset across different capture conditions (raindrop-focused, background-focused, and ground truth) for each of the 11 scenes.

| Scene | Raindrop-focused | Background-focused | GT |
|---|---|---|---|
| corner | 32 | 32 | 32 |
| beartoy | 29 | 29 | 29 |
| bicycle | 31 | 31 | 31 |
| dustbin | 30 | 30 | 30 |
| flover | 24 | 24 | 24 |
| parkbear | 28 | 28 | 28 |
| popmart | 31 | 31 | 31 |
| rustdesk | 28 | 28 | 28 |
| siyuanstone | 53 | 53 | 53 |
| train | 0 | 32 | 32 |
| yingjitongdao | 32 | 32 | 32 |
| **Total** | 318 | 350 | 350 |

Table 7: Quantitative analysis of the impact of camera pose estimation and point cloud initialization using COLMAP Schonberger & Frahm (2016b) and VGGT Wang et al. (2025) on 3DGS raindrop reconstruction performance. COLMAP failed entirely on the raindrop-focused dataset, and thus only results on the background-focused dataset are reported in the table. Diagonal entries in the table indicate reconstruction failures due to unsuccessful camera pose estimation. Notably, in the cases where COLMAP succeeded, it outperformed VGGT.

| | Scene | VGGT (3DGS) | | | COLMAP (3DGS) | | | VGGT (GS-W) | | | COLMAP (GS-W) | | | VGGT (WeatherGS) | | | COLMAP (WeatherGS) | | |
|---|---|---|---|---|---|---|---|---|---|---|---|---|---|---|---|---|---|---|---|
| | | PSNR | SSIM | LPIPS | PSNR | SSIM | LPIPS | PSNR | SSIM | LPIPS | PSNR | SSIM | LPIPS | PSNR | SSIM | LPIPS | PSNR | SSIM | LPIPS |
| BG-focused | corner | 17.458 | 0.453 | 0.453 | / | / | / | 19.253 | 0.561 | 0.387 | / | / | / | 17.282 | 0.392 | 0.413 | / | / | / |
| | beartoy | 17.249 | 0.625 | 0.435 | 21.452 | 0.785 | 0.343 | 19.474 | 0.711 | 0.415 | 18.623 | 0.688 | 0.451 | 15.611 | 0.584 | 0.442 | 19.993 | 0.723 | 0.332 |
| | bicycle | 19.670 | 0.456 | 0.353 | 20.822 | 0.647 | 0.253 | 19.722 | 0.496 | 0.361 | 17.864 | 0.443 | 0.559 | 19.278 | 0.404 | 0.328 | 20.227 | 0.512 | 0.280 |
| | dustbin | 16.780 | 0.374 | 0.568 | 19.414 | 0.516 | 0.431 | 19.205 | 0.484 | 0.691 | / | / | / | 17.069 | 0.347 | 0.467 | / | / | / |
| | flover | 14.272 | 0.188 | 0.578 | 16.649 | 0.336 | 0.464 | 14.814 | 0.283 | 0.667 | / | / | / | 9.977 | 0.098 | 0.723 | 10.710 | 0.118 | 0.726 |
| | parkbear | 17.479 | 0.435 | 0.464 | / | / | / | 18.953 | 0.507 | 0.490 | / | / | / | 17.271 | 0.376 | 0.441 | / | / | / |
| | popmart | 17.404 | 0.631 | 0.506 | 16.376 | 0.665 | 0.473 | 17.632 | 0.714 | 0.500 | 15.776 | 0.680 | 0.608 | 15.995 | 0.582 | 0.496 | 15.202 | 0.598 | 0.459 |
| | rustydesk | 21.255 | 0.484 | 0.429 | 21.741 | 0.615 | 0.377 | 21.498 | 0.543 | 0.499 | 17.813 | 0.500 | 0.704 | 20.721 | 0.419 | 0.365 | 21.726 | 0.539 | 0.322 |
| | siyuanstone | 20.538 | 0.552 | 0.402 | 20.128 | 0.659 | 0.349 | 21.688 | 0.603 | 0.439 | 21.620 | 0.627 | 0.489 | 21.043 | 0.564 | 0.307 | 19.990 | 0.586 | 0.334 |
| | yingjityongdao | 16.957 | 0.582 | 0.406 | / | / | / | 18.986 | 0.650 | 0.431 | / | / | / | 16.997 | 0.519 | 0.380 | / | / | / |

Frahm (2016b) successfully provided initialization information for 3DGS raindrop reconstruction, resulting in more detailed information.

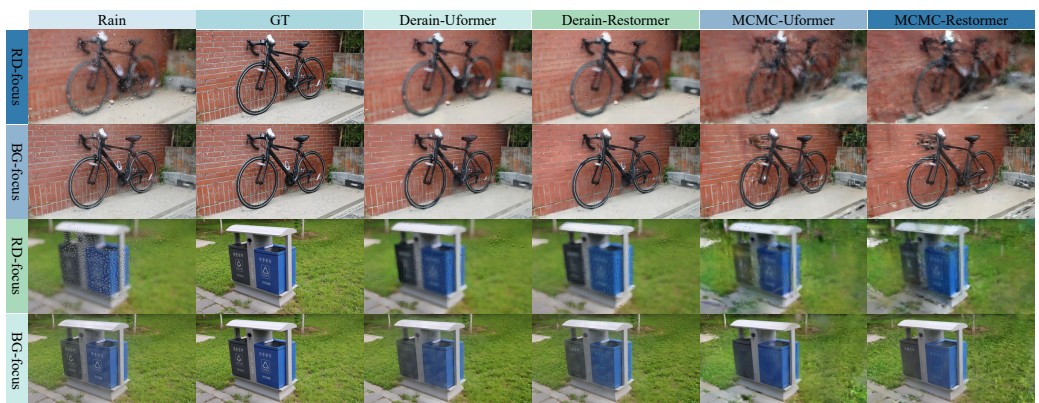

Figure 5: Qualitative comparison of single-image rain removal algorithms for 3DGS raindrop reconstruction. There is no significant difference between Uformer Wang et al. (2022) and Restormer Zamir et al. (2022).

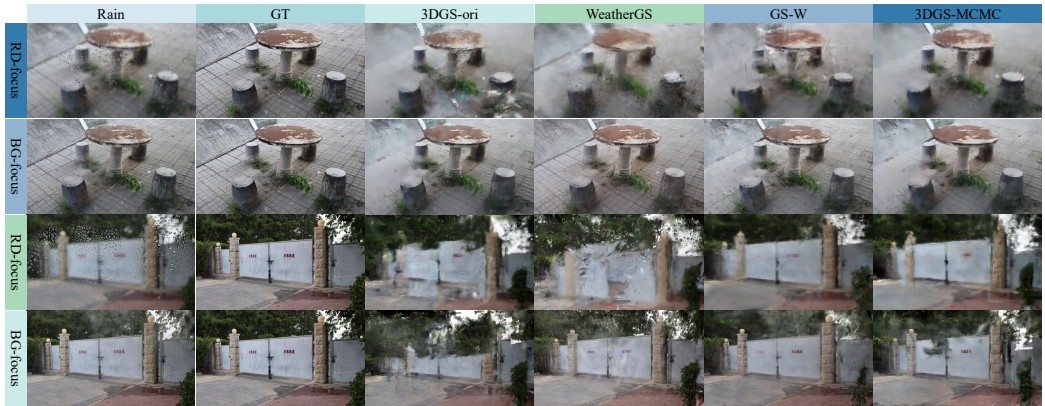

Figure 6: Qualitative comparative analysis of four 3DGS methods. Background-focused dataset are superior to raindrop-focused dataset because they provide more image details and higher reconstruction quality. WeatherGS Qian et al. (2024) has the best restoration quality for "blurred" images, while GS-W Zhang et al. (2024) performs best in overall restoration quality.

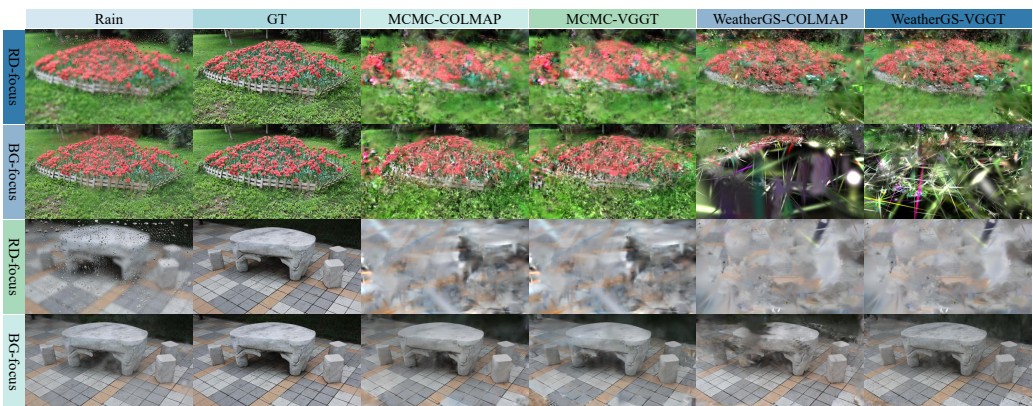

Figure 7: Qualitative analysis of the performance of COLMAP Schonberger & Frahm (2016b) and VGGT Wang et al. (2025) in 3DGS raindrop reconstruction. In successful cases, the use of COLMAP Schonberger & Frahm (2016b) enabled 3DGS methods to obtain more details.

Table 8: Quantitative comparison of the four methods: 3DGS Kerbl et al. (2023), WeatherGS Qian et al. (2024), GS-W Zhang et al. (2024), and 3DGS-MCMC Kheradmand et al. (2024). 3DGS-MCMC Kheradmand et al. (2024) achieves the best performance on the raindrop-focused dataset, while GS-W Zhang et al. (2024) performs best on the background-focused dataset.

| | Scene | 3DGS | | | WeatherGS | | | GS-W | | | 3DGS-MCMC | | |
|---|---|---|---|---|---|---|---|---|---|---|---|---|---|
| | | PSNR | SSIM | LPIPS | PSNR | SSIM | LPIPS | PSNR | SSIM | LPIPS | PSNR | SSIM | LPIPS |
| RD-focused | corner | 13.420 | 0.341 | 0.650 | 13.642 | 0.336 | 0.639 | 15.594 | 0.535 | 0.903 | 13.839 | 0.361 | 0.663 |
| | beartoy | 11.616 | 0.415 | 0.629 | 10.929 | 0.365 | 0.643 | 14.053 | 0.607 | 0.729 | 13.525 | 0.499 | 0.599 |
| | bicycle | 13.586 | 0.259 | 0.683 | 13.583 | 0.252 | 0.675 | 15.755 | 0.429 | 0.897 | 14.894 | 0.287 | 0.703 |
| | dustbin | 15.770 | 0.325 | 0.670 | 14.744 | 0.267 | 0.690 | 18.032 | 0.472 | 0.800 | 17.818 | 0.359 | 0.626 |
| | flover | 13.121 | 0.145 | 0.706 | 9.977 | 0.098 | 0.723 | 13.247 | 0.269 | 0.847 | 12.815 | 0.153 | 0.715 |
| | parkbear | 14.301 | 0.306 | 0.649 | 12.987 | 0.267 | 0.655 | 15.838 | 0.475 | 0.809 | 14.843 | 0.330 | 0.662 |
| | popmart | 13.271 | 0.520 | 0.593 | 12.234 | 0.461 | 0.609 | 15.691 | 0.693 | 0.648 | 15.276 | 0.602 | 0.562 |
| | rustydesk | 16.927 | 0.367 | 0.659 | 16.209 | 0.351 | 0.651 | 18.463 | 0.514 | 0.811 | 18.572 | 0.385 | 0.665 |
| | siyuanstone | 12.665 | 0.367 | 0.702 | 12.647 | 0.340 | 0.675 | 16.620 | 0.540 | 0.891 | 13.473 | 0.387 | 0.722 |
| | yingjityongdao | 14.261 | 0.416 | 0.624 | 13.748 | 0.340 | 0.617 | 17.701 | 0.589 | 0.744 | 16.039 | 0.461 | 0.625 |
| BG-focused | corner | 17.458 | 0.453 | 0.453 | 17.282 | 0.392 | 0.413 | 19.253 | 0.561 | 0.387 | 17.928 | 0.470 | 0.465 |
| | beartoy | 17.249 | 0.625 | 0.435 | 15.611 | 0.584 | 0.442 | 19.474 | 0.711 | 0.415 | 18.888 | 0.662 | 0.421 |
| | bicycle | 19.670 | 0.456 | 0.353 | 19.278 | 0.404 | 0.328 | 19.722 | 0.496 | 0.361 | 18.777 | 0.428 | 0.436 |
| | dustbin | 16.780 | 0.374 | 0.568 | 17.069 | 0.347 | 0.467 | 19.205 | 0.484 | 0.691 | 19.449 | 0.431 | 0.483 |
| | flover | 14.272 | 0.188 | 0.578 | 9.977 | 0.098 | 0.723 | 14.814 | 0.283 | 0.667 | 12.868 | 0.153 | 0.654 |
| | parkbear | 17.479 | 0.435 | 0.464 | 17.271 | 0.376 | 0.441 | 18.953 | 0.507 | 0.490 | 17.609 | 0.417 | 0.522 |
| | popmart | 17.404 | 0.631 | 0.506 | 15.995 | 0.582 | 0.496 | 17.632 | 0.714 | 0.500 | 17.894 | 0.683 | 0.457 |
| | rustydesk | 21.255 | 0.484 | 0.429 | 20.721 | 0.419 | 0.365 | 21.498 | 0.543 | 0.499 | 19.695 | 0.423 | 0.564 |
| | siyuanstone | 20.538 | 0.552 | 0.402 | 21.043 | 0.564 | 0.307 | 21.688 | 0.603 | 0.439 | 20.482 | 0.545 | 0.429 |
| | yingjityongdao | 16.957 | 0.582 | 0.406 | 16.997 | 0.519 | 0.380 | 18.986 | 0.650 | 0.431 | 18.598 | 0.603 | 0.427 |

Table 9: Quantitative analysis of the performance of two single-image restoration methods, Uformer Wang et al. (2022) and Restormer Zamir et al. (2022), applied to 3DGS methods. Uformer Wang et al. (2022) achieved slightly better performance than Restormer Zamir et al. (2022) in enhancing image quality for 3D reconstruction.

| | Scene | Uform. (3DGS) | | | Restorm. (3DGS) | | | Uform. (GS-W) | | | Restorm. (GS-W) | | | Uform. (GS-MCMC) | | | Restorm. (GS-MCMC) | | |
|---|---|---|---|---|---|---|---|---|---|---|---|---|---|---|---|---|---|---|---|
| | | PSNR | SSIM | LPIPS | PSNR | SSIM | LPIPS | PSNR | SSIM | LPIPS | PSNR | SSIM | LPIPS | PSNR | SSIM | LPIPS | PSNR | SSIM | LPIPS |
| RD-focused | corner | 13.420 | 0.341 | 0.650 | 13.170 | 0.332 | 0.653 | 15.594 | 0.535 | 0.903 | 15.673 | 0.532 | 0.890 | 13.839 | 0.361 | 0.663 | 13.709 | 0.359 | 0.663 |
| | beartoy | 11.616 | 0.415 | 0.629 | 11.535 | 0.428 | 0.623 | 14.053 | 0.607 | 0.729 | 14.103 | 0.597 | 0.743 | 13.525 | 0.499 | 0.599 | 12.976 | 0.484 | 0.605 |
| | bicycle | 13.586 | 0.259 | 0.683 | 13.469 | 0.248 | 0.671 | 15.755 | 0.429 | 0.897 | 15.314 | 0.428 | 0.904 | 14.894 | 0.287 | 0.703 | 14.755 | 0.283 | 0.686 |
| | dustbin | 15.770 | 0.325 | 0.670 | 15.673 | 0.319 | 0.667 | 18.032 | 0.472 | 0.800 | 18.120 | 0.472 | 0.793 | 17.818 | 0.359 | 0.626 | 17.566 | 0.357 | 0.617 |
| | flover | 13.121 | 0.145 | 0.706 | 12.972 | 0.143 | 0.712 | 14.814 | 0.283 | 0.667 | 7.013 | 0.002 | 1.066 | 12.815 | 0.153 | 0.715 | 12.708 | 0.152 | 0.715 |
| | parkbear | 14.301 | 0.306 | 0.649 | 14.246 | 0.301 | 0.648 | 15.838 | 0.475 | 0.809 | 16.013 | 0.479 | 0.813 | 14.843 | 0.330 | 0.662 | 15.176 | 0.336 | 0.653 |
| | popmart | 13.271 | 0.520 | 0.593 | 13.535 | 0.536 | 0.585 | 15.691 | 0.693 | 0.648 | 16.119 | 0.695 | 0.647 | 15.276 | 0.602 | 0.562 | 15.095 | 0.605 | 0.559 |
| | rustydesk | 16.927 | 0.367 | 0.659 | 17.251 | 0.367 | 0.655 | 18.463 | 0.514 | 0.811 | 18.219 | 0.515 | 0.800 | 18.572 | 0.385 | 0.665 | 18.373 | 0.382 | 0.662 |
| | siyuanstone | 12.665 | 0.367 | 0.702 | 12.707 | 0.368 | 0.697 | 19.620 | 0.540 | 0.891 | 15.997 | 0.533 | 0.895 | 13.473 | 0.387 | 0.722 | 13.732 | 0.385 | 0.711 |
| | yingjityongdao | 14.261 | 0.416 | 0.624 | 17.379 | 0.588 | 0.391 | 17.701 | 0.589 | 0.744 | 17.425 | 0.585 | 0.733 | 16.039 | 0.461 | 0.625 | 15.957 | 0.456 | 0.621 |
| BG-focused | corner | 17.458 | 0.453 | 0.453 | 17.570 | 0.450 | 0.442 | 19.253 | 0.561 | 0.387 | 19.101 | 0.558 | 0.366 | 17.928 | 0.470 | 0.465 | 17.944 | 0.477 | 0.452 |
| | beartoy | 17.249 | 0.625 | 0.435 | 17.128 | 0.609 | 0.449 | 19.474 | 0.711 | 0.415 | 19.755 | 0.701 | 0.426 | 18.888 | 0.662 | 0.421 | 18.877 | 0.651 | 0.430 |
| | bicycle | 19.670 | 0.456 | 0.353 | 19.379 | 0.447 | 0.373 | 19.722 | 0.496 | 0.361 | 19.412 | 0.486 | 0.402 | 18.777 | 0.428 | 0.436 | 18.615 | 0.425 | 0.451 |
| | dustbin | 16.780 | 0.374 | 0.568 | 16.258 | 0.358 | 0.548 | 19.205 | 0.484 | 0.691 | 19.701 | 0.487 | 0.651 | 19.449 | 0.431 | 0.483 | 19.194 | 0.428 | 0.459 |
| | flover | 14.272 | 0.188 | 0.578 | 14.154 | 0.175 | 0.582 | 14.814 | 0.283 | 0.667 | 14.362 | 0.275 | 0.686 | 12.868 | 0.153 | 0.654 | 12.832 | 0.151 | 0.649 |
| | parkbear | 17.479 | 0.435 | 0.464 | 17.513 | 0.431 | 0.449 | 18.953 | 0.507 | 0.490 | 18.931 | 0.504 | 0.473 | 17.609 | 0.417 | 0.522 | 17.507 | 0.407 | 0.520 |
| | popmart | 17.404 | 0.631 | 0.506 | 16.351 | 0.611 | 0.513 | 17.632 | 0.714 | 0.500 | 17.451 | 0.709 | 0.518 | 17.894 | 0.683 | 0.457 | 18.645 | 0.682 | 0.452 |
| | rustydesk | 21.255 | 0.484 | 0.429 | 20.854 | 0.476 | 0.402 | 21.498 | 0.543 | 0.499 | 21.345 | 0.532 | 0.395 | 19.695 | 0.423 | 0.564 | 19.316 | 0.414 | 0.551 |
| | siyuanstone | 20.538 | 0.552 | 0.402 | 20.824 | 0.546 | 0.397 | 21.688 | 0.603 | 0.439 | 21.572 | 0.597 | 0.442 | 20.482 | 0.545 | 0.429 | 20.179 | 0.538 | 0.438 |
| | yingjityongdao | 16.957 | 0.582 | 0.406 | 17.379 | 0.588 | 0.391 | 18.986 | 0.650 | 0.431 | 19.115 | 0.639 | 0.432 | 18.598 | 0.603 | 0.427 | 18.377 | 0.601 | 0.423 |

