# OpenReview forum: "Raindrop GS: A Benchmark for 3D Gaussian Splatting under Raindrop Conditions"
_ICLR.cc/2026/Conference — ICLR 2026 Conference Withdrawn Submission_

### Official Review · Reviewer_Mnyg · 2025-10-26

**Soundness:** 2
**Presentation:** 2
**Contribution:** 2
**Rating:** 2
**Confidence:** 5

**Summary:**

This paper introduces RaindropGS, the comprehensive benchmark designed to evaluate the full 3D Gaussian Splatting (3DGS) pipeline under raindrop-contaminated conditions. The benchmark spans data acquisition, camera pose estimation, point cloud initialization, single-image raindrop removal, and 3DGS reconstruction. A real-world dataset is constructed with three aligned image sets per scene: raindrop-focused, background-focused, and rain-free ground truth. Through extensive experiments and analysis, the paper reveals critical performance limitations of existing 3DGS models under realistic raindrop interference and identifies key challenges for future research.

**Strengths:**

1. The dataset faithfully captures optical distortions induced by raindrops with diverse appearances and focus variations, providing higher realism than synthetic datasets.
2. The benchmark establishes a standardized platform that can guide future research toward improving 3DGS robustness in adverse weather conditions.
3. The paper is the first to evaluate the entire 3DGS pipeline under real-world raindrop contamination, addressing a critical and underexplored problem.

**Weaknesses:**

1. The paper mainly presents a benchmark rather than introducing new algorithmic advances or explicit solutions to raindrop interference.
2. While PSNR and SSIM are reported, the benchmark does not incorporate uncertainty-aware or 3D consistency metrics to better assess robustness.
3. The qualitative analysis is limited, and failure cases are not extensively visualized to illustrate differences between methods in detail preservation and artifact handling.
4. The benchmark does not explore its applicability to other degradation conditions such as fog or snow, which may reduce its broader impact.
5. Despite emphasizing the impact of raindrops on reconstruction quality, the paper does not introduce any baseline specifically designed to model raindrop geometry or optical distortion, making it difficult to assess how much room there is for methodological improvement.
6. The benchmark relies on various external tools and pretrained models with non-standardized parameter settings, and lacks a unified error propagation protocol, which may lead to significant reproducibility issues across different users.

**Questions:**

See the above parts.

---

### Official Review · Reviewer_o6pu · 2025-10-31

**Soundness:** 2
**Presentation:** 3
**Contribution:** 2
**Rating:** 4
**Confidence:** 4

**Summary:**

This paper introduces Raindrop-GS, a benchmark dataset designed to study the impact of raindrop degradation on 3D scene reconstruction. It provides paired raindrop-focused and background-focused images with corresponding clean ground truth, enabling controlled evaluation of reconstruction quality under adverse weather conditions. The authors benchmark several representative 3D methods such as COLMAP and VGGT, showing that raindrops cause significant degradation in geometric accuracy.

**Strengths:**

- The paper is well structured and easy to follow.

- The authors collect paired raindrop-focused and background-focused images for each scene, offering new dataset for the study of optical distortion and occlusion effects in 3D restoration and reconstruction. I genuinely appreciate the effort invested in data acquisition, as capturing aligned multi-focus image pairs under real-world conditions is both technically challenging and time-consuming.

**Weaknesses:**

- While collecting 3D datasets under adverse weather conditions is undoubtedly challenging, this work focuses solely on raindrop degradation. Other rain-related degradations such as rain streaks, rain fog, and surface scattering are not included. As a result, the dataset does not comprehensively represent the wide variety of rainy conditions encountered in real-world outdoor scenes. Moreover, the raindrop setting itself should also consider dual-focus capture systems (e.g., the CVPR NTIRE Raindrop Challenge: https://lixinustc.github.io/CVPR-NTIRE2025-RainDrop-Competition.github.io/). Therefore, the dataset type is somewhat narrow and lacks reference to other established degradation setups in the rain domain.

- Given its modest size and limited diversity, the analysis conducted in the paper remains relatively constrained. The authors mainly compare COLMAP and VGGT on point-cloud reconstruction accuracy, which provides only a narrow view of performance differences. However, rain degradation in practice affects many other factors such as feature extraction stability, pose accuracy, and surface consistency. Simply stating that “there are substantial gaps” between clean and raindrop conditions does not provide sufficient insight into the underlying causes or possible mitigation strategies.

- The paper is a pure dataset paper and introduce no novel method.

**Questions:**

Please see the weakness section.

---

### Official Review · Reviewer_EJGy · 2025-10-31

**Soundness:** 2
**Presentation:** 3
**Contribution:** 2
**Rating:** 2
**Confidence:** 4

**Summary:**

The paper introduces a benchmark for evaluating 3D Gaussian Splatting (3DGS) methods under raindrop conditions. Specifically, the authors capture a real-world dataset with different focus settings. They then analyze the impact of various components on the reconstruction quality of 3DGS-based methods in raindrop scenarios, including two commonly used camera pose estimation and point cloud initialization methods, three raindrop removal algorithms, and the effects of different reconstruction strategies employing several 3DGS variants. The experimental results provide valuable insights and have positive implications for existing 3D reconstruction approaches in raindrop-affected environments

**Strengths:**

1. The paper constructs a real-world raindrop dataset specifically designed for evaluating 3D reconstruction methods under challenging rain-contaminated conditions. Compared to existing synthetic datasets, it offers a unique combination of raindrop-focused, background-focused, and clear ground-truth image sets, facilitating a comprehensive evaluation of the effects of different camera focal settings. Moreover, the variability of raindrops at each viewpoint—random in shape, number, and size—closely mimics real-world conditions, enabling a more realistic and robust assessment of 3DGS-based reconstruction methods.
2. The paper tests two widely adopted methods for camera pose estimation and point cloud initialization—COLMAP and VGGT—in raindrop-affected environments. This analysis provides valuable insights into the limitations and challenges faced by existing 3D reconstruction pipelines when operating under adverse weather conditions.
3. By evaluating three single-image raindrop removal methods within the pipeline, the paper offers important perspectives on how different deraining techniques impact overall reconstruction quality.
4. The paper also evaluates several 3DGS variants to assess their robustness and effectiveness in raindrop-affected scenes. Overall, the paper provides a relatively comprehensive benchmark that advances the evaluation of raindrop-conditioned 3D reconstruction methods.

**Weaknesses:**

1. Although the paper presents a dataset containing both raindrop-focused and background-focused conditions, it lacks a clear evaluation of the impact of different raindrop intensities. Additionally, some scenes in the dataset encounter initialization failures, but the paper does not provide an in-depth analysis of these issues.
2. The paper only evaluates three single-image raindrop removal methods and lacks a discussion on multi-view consistency and its influence on 3D reconstruction.
3. Overall, the work lacks novelty; the proposed pipeline is based entirely on existing mature methods and is constrained by current technological bottlenecks. Moreover, the overall process closely follows the classic 3DGS pipeline without proposing targeted improvements or innovative methods specifically designed for reconstruction tasks in raindrop scenarios.
4. In raindrop-focused scenes, some samples fail to initialize point clouds properly, for which the paper employs a random point cloud initialization strategy. However, the stability of this random initialization and its impact on the final reconstruction quality are not discussed.

**Questions:**

1. There appear to be inconsistencies between the results reported in Table 4 and Table 5, particularly regarding the GS-W method with VGGT and Uformer preprocessing on the background-focused dataset. Could the authors clarify the reasons behind these discrepancies?
2. The comparative analysis of different single-image raindrop removal methods, especially as presented in Table 3, lacks sufficient detail. It is unclear under which initialization and 3D reconstruction methods these deraining results were obtained. Could the authors please clarify?
3. Table 2 shows that VGGT yields more accurate camera pose estimates compared to COLMAP. However, the original 3DGS method achieves the best reconstruction performance when using the COLMAP results. Could the authors explain this apparent contradiction?
4. For raindrop-focused scenes where standard initialization fails, a random point-cloud initialization strategy is employed. However, the manuscript does not provide an in-depth discussion about its stability or impact on the final reconstruction quality. Could the authors elaborate on these aspects?

---

### Official Review · Reviewer_yCRt · 2025-10-31

**Soundness:** 3
**Presentation:** 3
**Contribution:** 2
**Rating:** 4
**Confidence:** 5

**Summary:**

The paper introduces a new dataset for raindrop removal then 3D reconstruction. The dataset is 11 scenes with ~38 multi-views and sub-categorized into raindrop focused and background focused. The paper then evaluates existing deraining methods and/or downstream 3DGS frameworks on this dataset.

**Strengths:**

1.  The authors are right in identifying that photography is not only done with the background in focus under rainy conditions but also with the droplets in focus. This problem setting identification is unique and present in the dataset introduced.
2. The paper is well written and clearly presented.

**Weaknesses:**

1. The dataset introduced, in my opinion, is small (11 scenes; same as DerainGS) to be justified as a benchmark for the deraining + 3D reconstruction task.
2. The paper claims domain shift between simulations to real world de-raining training outputs but shows no such results.
3. There is no figure showing all the 11 scenes together. (To understand the diversity in them)
4. Evaluation on the dataset just reveals which existing methods work or not. The analysis does not point what might be the contributing factors or what in the introduced dataset causes certain methods to fail or work. This contextualization or positioning the evaluation against the dataset would strengthen the paper.
5. Supplementary shows no methods work on raindrop focused scenes. Any starting leads or preliminary insights for reconstruction would be nice to have (but not necessary) in the discussions section.

**Questions:**

See weaknesses point 2, 3 and 4.
Providing these would strengthen the paper.

Additionally, would it be possible to show the capturing setup or make a diagram of the same?

---

### Note · Authors · 2025-11-12

I have read and agree with the venue's withdrawal policy on behalf of myself and my co-authors.